# Modulatory Effects of Autophagy on APP Processing as a Potential Treatment Target for Alzheimer’s Disease

**DOI:** 10.3390/biomedicines9010005

**Published:** 2020-12-24

**Authors:** Md. Ataur Rahman, Md Saidur Rahman, MD. Hasanur Rahman, Mohammad Rasheduzzaman, ANM Mamun-Or-Rashid, Md Jamal Uddin, Md Rezanur Rahman, Hongik Hwang, Myung-Geol Pang, Hyewhon Rhim

**Affiliations:** 1Center for Neuroscience, Brain Science Institute, Korea Institute of Science and Technology (KIST), Seoul 02792, Korea; hongik.kist@gmail.com; 2Global Biotechnology & Biomedical Research Network (GBBRN), Department of Biotechnology and Genetic Engineering, Faculty of Biological Sciences, Islamic University, Kushtia 7003, Bangladesh; 3Department of Animal Science & Technology and BET Research Institute, Chung-Ang University, Anseong 456-756, Korea; shohagvet@gmail.com (M.S.R.); mgpang@cau.ac.kr (M.-G.P.); 4Department of Biotechnology and Genetic Engineering, Bangabandhu Sheikh Mujibur Rahman Science and Technology University, Gopalganj 8100, Bangladesh; hasanurrahman.bge@gmail.com; 5School of Biomedical Sciences, Institute of Health and Biomedical Innovation, Queensland University of Technology, Brisbane, QLD 4059, Australia; m.rasheduzzaman@qut.edu.au; 6Anti-Aging Medical Research Center and Glycation Stress Research Center, Graduate School of Life and Medical Sciences, Doshisha University, Kyoto 602-8566, Japan; mamunbtgeiu@gmail.com; 7Graduate School of Pharmaceutical Sciences, College of Pharmacy, Ewha Womans University, Seoul 03760, Korea; hasan800920@gmail.com; 8ABEx Bio-Research Center, East Azampur, Dhaka 1230, Bangladesh; 9Department of Biochemistry and Biotechnology, School of Biomedical Science, Khwaja Yunus Ali University, Sirajgonj 6751, Bangladesh; rezanur12@yahoo.com; 10Division of Bio-Medical Science and Technology, KIST School, Korea University of Science and Technology (UST), Seoul 02792, Korea

**Keywords:** autophagy, amyloid precursor protein (APP), β-amyloid (Aβ), mTOR, Alzheimer’s disease (AD)

## Abstract

Alzheimer’s disease (AD) is characterized by the formation of intracellular aggregate composed of heavily phosphorylated tau protein and extracellular deposit of amyloid-β (Aβ) plaques derived from proteolysis cleavage of amyloid precursor protein (APP). Autophagy refers to the lysosomal-mediated degradation of cytoplasmic constituents, which plays a critical role in maintaining cellular homeostasis. Importantly, recent studies reported that dysregulation of autophagy is associated in the pathogenesis of AD, and therefore, autophagy modulation has gained attention as a promising approach to treat AD pathogenesis. In AD, both the maturation of autolysosomes and its retrograde transports have been obstructed, which causes the accumulation of autophagic vacuoles and eventually leads to degenerating and dystrophic neurites function. However, the mechanism of autophagy modulation in APP processing and its pathogenesis have not yet been fully elucidated in AD. In the early stage of AD, APP processing and Aβ accumulation-mediated autophagy facilitate the removal of toxic protein aggregates via mTOR-dependent and -independent pathways. In addition, a number of autophagy-related genes (Atg) and APP are thought to influence the development of AD, providing a bidirectional link between autophagy and AD pathology. In this review, we summarized the current observations related to autophagy regulation and APP processing in AD, focusing on their modulation associated with the AD progression. Moreover, we emphasizes the application of small molecules and natural compounds to modulate autophagy for the removal and clearance of APP and Aβ deposits in the pathological condition of AD.

## 1. Introduction

Alzheimer’s disease (AD) is the most irreversible and progressive brain disorder of neurodegenerative disease characterized by the accumulation of extracellular amyloid beta (Aβ) leading to the formation of senile plaques and intracellular tau aggregates that form neurofibrillary tangles (NFTs) in the brain [1,2]. Approximately, 70% of AD risk is considered to be inherited and numerous genes are frequently involved, although the actual cause and molecular mechanisms are poorly understood [2,3,4]. Amyloid precursor protein (APP), a transmembrane glycoprotein (type I), is the key molecular driver of AD pathogenesis. An extracellular domain and a small cytosolic domain present in APP are generally accepted to be responsible for AD progression [5]. APP is ubiquitously present in the brain and is involved in building synaptic network as well as regulating neurogenesis [6]. In addition, APP has a modulatory effect on cell surface receptors and axonal transport. However, the exact functionality of APP still remains elusive [7]. In general, upon its synthesis in the endoplasmic reticulum, APP undergoes phosphorylation and glycosylation and is finally transported into the Golgi apparatus. Additional processing of APP occurs in the trans-Golgi-network (TGN), and the highest concentrations of APP are found in the TGN under normal physiological conditions. Cleavage of APP by α-secretase produces a soluble molecule, sAPPα, within the Aβ domain [8], and APP is taken up as a cargo by the endosomal/lysosomal degradation pathway. Lysosomal degradation is a clearance mechanism required to maintain a healthy state and to prevent the accumulation of undesirable cellular waste materials. The generated peptides play an important role in synaptic plasticity and neuronal survival in the healthy state [9]. It has been reported that early-onset AD (EOAD) is usually inherited with certain autosomal dominant alleles, however, in late-onset AD (LOAD) such inheritance is unknown. While individuals bearing one inherited copy of the APOE e4 allele have a great risk of developing AD, people who inherit two copies have a greater risk of AD [10]. In some instances, EOAD is triggered through genetic mutations that are passed on from parent to child, which is usually known as early onset familial AD (FAD). Generally, FAD occurs due to mutations in presenilin 1 (PSEN1), presenilin 2 (PSEN2), and APP genes through β-secretase (BACE-1) and γ-secretase instead of α-secretase leads to unwanted assembly and accumulation of Aβ peptides in the brain, [11], thereby causing AD pathogenesis [12]. Diffusible oligomers and insoluble senile plaques are formed due to the abnormal presentation of Aβ peptides, thereby resulting in higher neurotoxicity. Moreover, fibrillary plaques are found in the intracellular spaces due to the aggregation of abnormal Aβ oligomers [13]. Hyperphosphorylation of tau, aggregation of hyperphosphorylated tau to bind and stabilize microtubules, is related to its aggregation as well as the formation of neurofibrillary tangles (NFTs) and is considered as a pathological condition of AD [14]. NFTs are formed by intracellular aggregation of hyperphosphorylated tau protein in specific brain regions [15]. Collectively, the neurotoxic effects of NFTs and Aβ associated with the excessive accumulation of extracellular plaques in the brain are a hallmark of AD pathogenesis [16].

Autophagy plays a widespread role in both physiological and multiple pathological conditions, including cancer and neuronal disorders [17], and is extensively involved in the pathogenesis of AD [18]. Autophagy, a self-digesting mechanism, is an intracellular cleansing process characterized by the engulfment of malformed proteins and damaged cellular organelles by membrane-bound vesicles known as autophagosomes [17]. These autophagosomes subsequently fuse with the lysosomes to form autolysosomes resulting degradation of dysfunctional materials by lysosomal acid hydrolases [19]. Autophagy is a complex and tightly regulated enzymatic process that is largely classified into two categories: mammalian target of rapamycin (mTOR)-independent and -dependent autophagy. A defect in the autophagy-lysosomal pathway has been linked to AD, which induces the formation of toxic Aβ aggregates and causes cellular apoptosis as well as tissue and organ damage, culminating in clinical symptoms [20]. In the initial stages of AD, Aβ may induce autophagy to accelerate their removal process by employing both mTOR-independent and -dependent pathways. Progression of AD deregulates the autophagy pathway, resulting in the continuous generation of Aβ, which exaggerates both autophagy malfunction and AD [21]. In addition, both oligonucleotides and proteins, such as miRNAs, transcription factor EB (TFEB), PSEN1, Nrf2, and Beclin-1 are simultaneously impaired in the regulation of autophagy, which are meticulously interrelated in the pathogenesis of AD [22]. Therefore, it is evident that the regulation of autophagy is crucial for APP clearance and the inhibition of AD pathogenesis. Abnormal autophagy is associated with AD pathogenesis; therefore, targeting autophagy may have a profound role in AD management [23].

## 2. Autophagy Pathway

The autophagy pathway is initiated by the generation of the phagophore and terminates upon the clearance of the autophagosomal cargo. Degradation of the autophagosomal cargo results in recycling of the cargo [24]. The Atg12-Atg5-Atg16 complex and microtubule-associated protein light chain 3 (LC3-I) and its phospholipid conjugate (LC3-II), control autophagosome formation. Both LC3-I and II are used as markers for the double-membrane vesicles, autophagosomes. Furthermore, p62 is a marker for autophagic cargo degradation in the autolysosomes. Therefore, the levels of these markers can indicate the activation of autophagic flux [25].

Starvation initiates autophagy which is characterized by the formation of a double-membrane structure, known as the autophagosome, that sequesters cytoplasmic materials for degradation. Autophagosomes fuse with the lysosomes, and acid hydrolases present in the lysosomes degrade the cargo, which also contributes to mitochondrial quality control and cellular homeostasis [26]. The process of autophagy involves the following steps: the formation of an isolation membrane (nucleation), membrane elongation, autophagosome maturation, docking, and fusion of the autophagosome with the endosome and finally with the lysosome to form the autolysosome, and degradation of the internal materials inside the autolysosome [24] (Figure 1). Autophagy is controlled by numerous growth factors as well as nutrient signaling, such as mTOR complex 1 (mTORC1) and class I PI3K/Akt signaling pathways. Dysregulation of autophagy results in compromised recycling leading to the accumulation of unwanted debris within a cell; therefore, inhibition of autophagy is implicated in multiple pathophysiological processes, including neurodegenerative diseases such as AD [17].

### 2.1. mTOR-Dependent Autophagy Pathway

Mammalian target of rapamycin (mTOR) is an essential serine-threonine protein kinase, comprised of mTOR complex 1 (mTORC1) and mTOR complex 2 (mTORC2) [27]. mTOR is known as a classical regulator of autophagy and controls vital cellular functions, such as protein translation and cell growth [28,29]. mTOR activity is closely associated with numerous factors, including chronic stress, starvation, and glucocorticoids. During starvation, mTORC1 activity is downregulated, thereby initiating autophagy to recycle intracellular constituents and thus generate a source of energy. Phosphoinositide 3-kinase (PI3K) and protein kinase B (Akt/PKB) are molecules upstream of mTOR [27]. These two molecules interact with mTOR and modulate the PI3K/Akt/mTOR pathway that controls autophagy. However, blockage or inhibition of any of the molecules of this pathway triggers autophagy, thereby augmenting the clearance or removal of Aβ in AD [27]. Previous studies have shown that inhibition of mTORC1 during starvation or its pharmacological blockade using inhibitors such as rapamycin, CCI-779, Torin1, or PP242 stimulates autophagy [30,31].

Furthermore, adenosine 5′-monophosphate-activated protein kinase (AMPK) induces autophagy [29]. AMPK is an upstream regulator of mTOR. However, peroxisome proliferator-activated receptor-γ (PPARγ) and the PPARγ/AMPK/mTOR axis also regulate autophagy. A previous study has been suggested that dihydroceramide desaturases (Des1 and Des2) are enzymes that catalyze the synthesis of dihydroceramide into ceramide, which can trigger the levels of mTORC1 and thereby inhibit autophagy, although the underlying mechanism is not clear. Later, it has been found that Des1, can downregulate the levels of mTORC1 which in turned inhibition of p70S6K1 activity and finally promote autophagic activity (Figure 2), thereby suppressing amyloid secretion in AD [32]. In addition, the transient receptor potential mucolipin-1 (TRPML1) is extensively expressed in lysosomes, which also serves as an autophagy regulator. Inactivation of TRPLM1 as well as PPARγ/AMPK signaling pathway blocks mTOR signaling, thereby triggering the accumulation of degraded cellular components via inhibition of autophagy flux. A study using transgenic mice revealed that TRPML1 is a precursor for the progression of AD due to the blocking of autophagy machinery [33]. Besides, DNA damage, oxidative stress, hypoxia, and metabolic stress generate reactive oxygen species (ROS), which modulate autophagy via the Akt/mTOR pathway [34,35].

### 2.2. mTOR-Independent Autophagy Pathway

An increase in AMPK-mediated phosphorylation activates autophagy machinery. Ca^2+^-dependent protein kinase β (CaMKKβ) is an upstream controller of AMPK, and the influx of Ca^2+^ through TRPM7 maintains basal autophagy via the CaMKKβ/AMPK pathway. The aggregation of Aβ interferes with Ca^2+^ homeostasis, leading to mitochondrial dysfunction, which is closely connected to AD pathogenesis [36]. Generally, AMPK-mediated phosphorylation occurs at serine-317/777 sites of the autophagy initiation kinase ULK1. In addition, AMPK, a signaling molecule upstream of Beclin-1, is involved in the initiation of the pre-autophagosomal complex. And it directly phosphorylates the serine-91/94 sites of Beclin-1, thereby initiating autophagy (Figure 2). Inflammatory response activates the microglia, which increases the transportation of p-tau in neurons and assists the degradation of p-tau in lysosomes. This course of action increases autophagic flux in microglia and assists the clearance of cellular debris in a regular manner. ROS production by mitochondria causes oxidative damage to mitochondrial proteins and triggers autophagy-mediated cell death in an mTOR independent manner [37].

## 3. Neuronal Roles of APP

Biochemical as well as genetic evidence have established that APP plays a central role in AD pathophysiology mainly due to consecutive proteolytic cleavage that results in the formation of Aβ plaques [38]. Recent evidence has shown that APP is vital for the generation, differentiation, and migration of neurons [39]. Robust APP expression is able to rescue these neuronal cell phenotypes significantly. In addition, APP also plays a significant role in *Drosophila melanogaster*, and suppression of the APPL gene causes an alteration in chemotactic behavior [40]. Interestingly, high levels of APPL are associated with neuronal regeneration in a brain injury model of *Drosophila*, which increases mortality in APPL mutant flies. APPL is positively correlated with increased protrusions of dendritic neurites, and the potential role of APP in axonal growth after a traumatic brain injury has been previously reported [41]. Moreover, overexpression of APP promotes synaptic differentiation, and APPL mutation results in reduced number of synaptic lobes in the *Drosophila* neuromuscular junction [42].

It has been found that Aβ peptides are produced from intracellular organelles including ER, moreover found extracellularly. Additionally, it is distributed that subcellular localization and processing of amyloid-β protein precursor (AβPP) and the proteolytic products are found in mitochondria-associated membranes (MAMs) that might be contributed to AD pathogenesis [43]. However, AβPP is folded which further modified in ER in addition to transport via golgi complex to plasma membrane. It has been proposed that Aβ oligomers store in the ER lumen due to deficits in axonal transport [44]. Furthermore, it has been revealed that AβPP and its catabolites relate with MAMs which regulating mitochondria as well as ER functions [43]. Importantly, the physiological function of APP, both intracellular and extracellular domains, are required to mediate synaptogenic activity and synaptic dysfunction happening in AD pathogenesis. Moreover, APP proteins are essential in either neuronal survival, differentiation, synaptophysin transport vesicles to synaptic sites, axon degeneration and pruning, cell adhesion, and apoptosis [45].

APP is ubiquitously expressed in mammalian cells and has a multifunctional role in cellular functions, such as cell adhesion, differentiation of neuronal cells, nerve migration, synapse formation, and neurite growth. APP immunoreactivity was reported to increase after brain injury in mice, as seen during traumatic injury of the brain. APP-deficient mice showed weight loss, loss of balance and muscle weakness, impaired behavior, and long-term potentiation [46]. Evidence from other animal models of APP deficiency has demonstrated the potential role of APP in the generation, differentiation, and migration of neurons. The potentially important role of APP is part of a complex mechanism involved in several neurological functions, such as nerve development [47]. Growing evidence suggests that soluble APPα (sAPPα) plays a neuroprotective role and functions similar to growth factors, and it has been shown that the APP intracellular domain (AICD) interacts with numerous proteins involved in the regulation of transcription and axonal transport [48].

## 4. Proteolytic Processing of APP in Alzheimer’s Disease

APP is proteolytically cleaved into several fragments during intracellular transport, and these metabolites of APP mediate multiple cellular functions, some of which are harmful. Thus, the net effect of full-length APP on the activity of cells may be due to the combination of different metabolite roles that mainly depend on the percentage of each APP metabolite level. APP may undergo non-amyloidogenic or amyloidogenic excision by several secretases [49,50]. The details of APP processing by amyloidogenic and non-amyloidogenic process are shown in Figure 3.

The amyloidogenic processing of APP, which is first mediated by β-secretase, leads to the production of a large soluble amyloid precursor protein β (sAPP β) containing a carboxy-terminal fragment β (CTFβ) (C99) [50]. In the brain, the β-site cleavage enzyme (BACE1) that cleaves APP is considered as the primary β secretase. Furthermore, C99 fragments are cleaved by γ-secretase, which produces two fragments known as Aβ and carboxy-terminal fragment γ (CTFγ) or APP intracellular domain (AICD) [51]. Eventually, Aβ fragments generate Aβ oligomers which subsequently form fibrils resulting in the formation of Aβ plaques [52]. In the non-amyloidogenic pathway, APP undergoes proteolytic processing by an α-secretase complex containing presenilin [53]. Non-amyloidogenic processing produces an intracellular carboxy-terminal fragment (CTF83) and a soluble amyloid precursor protein α (sAPPα), which is thought to play a neuroprotective role in contrast to Aβ [54]. The CTF83 fragments are consecutively cleaved in the transmembrane domain by the action of γ-secretase to produce carboxy-terminal fragment γ (CTFγ) or AICD and P3 fragment [55,56]. In contrast, higher levels of α-secretase increase AICD43 generation [57]. Moreover, α-secretase activity is related to several members of the ADAM protein family such as ADAM17, ADAM10, and ADAM9, although additional proteases may also be involved [58]. Collectively, the cleavage of APP by α-secretase and β-secretase may have different effects on the subsequent release of AICD [51].

In addition, Aβ oligomers enhance the phosphorylation and aggregation of tau proteins. Moreover, tau aggregates and Aβ form small soluble oligomers as well as large insoluble fibrils in AD [59]. Evidence also suggests that suppression of the tau protein decreases the production of Aβ and inhibits the toxicity induced by the feedback mechanism [16]. On the other hand, phosphorylated tau causes destabilization of the microtubules, degeneration of the cell membrane, and intracellular aggregation of NFTs, which eventually leads to cell death [15]. In addition, it has also been proposed that Aβ accumulation and aggregation of p-tau causes ER stress and contributes to synaptic dysfunction as well as neurodegeneration in AD [16]. The ER regulates protein folding, modification, and quality control, and mild stress in the ER causes denaturation of misfolded and aggregated proteins in the ER lumen, prompting an unfolded protein response (UPR) to restore the homeostasis of proteins. However, excessive stress promotes UPR to trigger proapoptotic programs and cause cell death [60] (Figure 4).

## 5. Amyloid Precursor Protein (APP) Processing in Autophagy Pathway

Increasing evidence shows that the autophagy process is impaired in various neurodegenerative diseases, including AD. Aging also plays a role in dysfunction of the autophagy process, which ultimately leads to AD pathogenesis [61]. The activation of autophagy is associated with the reduction of Aβ deposition and improves memory deficits in AD mice. Autophagy is the main mechanism for altering the production of APP and intracellular Aβ peptide. Autophagic vacuoles (AVs) contain immunoreactive Aβ and its precursor proteins in AD models [62]. In addition, the levels of APP in AVs are constant, implying that the cleavage of APP occurs in the AVs [63]. This is supported by data indicating that such AVs contain secretase responsible for producing CTF [16]. In addition, AV fractions comprise notable levels of presenilin-1 in addition to nicastrin, demonstrating that AVs are the sites for abnormal APP cleavage [62] (Figure 5).

### 5.1. Autophagy and Aβ Processing

APP is cleaved by β-secretase (BACE1) and γ-secretase to yield Aβ [51,64]. Acceleration of AD progression was observed due to enhanced activity of BACE1 as well as γ-secretase, which increases APP processing and the formation of Aβ [65]. Autophagy plays a critical role in the processing of APP [66]. In animal models of AD, triggering Atg5-dependent autophagy stimulates early degradation of APP, and thus inhibits Aβ accumulation [67]. Inhibition of mTOR induces autophagy and decreases the expression levels of BACE1 in an APP/PS1 transgenic mouse model of AD [68]. Mutant APP was reported to impair energy metabolism in the mitochondria of AD neurons. In addition, dysfunction of autophagy activates γ-secretase and stimulates APP cleavage leading to the formation of Aβ. It has also been reported that unconventional autophagy activates APP cleavage, which leads to the production of Aβ. The autophagy inhibitor, 3-methyladenine (3-MA), enhances γ-secretase activity and stimulates Aβ production [69]. Several studies have indicated that APP cleavage results in Aβ production which is considered as an autophagy substrate [62], and therefore, maintaining normal autophagy is important for the removal and clearance of APP [70]. However, the molecular mechanism by which APP serves as a substrate for autophagy remains unclear.

### 5.2. Dysfunctional Autophagy and Aβ Processing

In the initial phase of AD, autophagy can be activated by Aβ formation, and Aβ is likely to be degraded by the autophagosome-lysosomal system [70]. Evidence suggests that Aβ is expressed in abnormal autophagic vesicles, which could be a source of extracellular Aβ plaque formation in a *Drosophila* model [71]. Autophagy may contribute to Aβ secretion through the secretory pathway or a secretory lysosomal pathway; similarly, absence of neuronal autophagy might reduce Aβ secretion. For this reason, autophagy is suggested to play a dual role in Aβ degradation as well as secretion. Therefore, additional studies are required to investigate the dual role of autophagy in the clearance and secretion of Aβ in the pathogenesis of AD [72]. In the later stage of AD, the continuous accumulation of Aβ induces abnormal autophagy, which leads to neuronal dysfunction and accelerates AD symptoms. The toxic form of Aβ, Aβ-derived diffusible ligands (ADLLs), is involved in the development of AD and regulates autophagy [73]. The exposure of neuronal cells to ADLLs decreased the phosphorylated levels of p70S6K1, indicating that mTOR inhibition governs the outcome in ADLL-mediated abnormal autophagy [73]. Aβ increases ROS generation resulting in hyperactivation of autophagy via NOX4 upregulation, leading to neuronal cell death [74]. Interestingly, reduction of NOX4 and ROS levels can prevent over-activation of autophagy as well as neuronal cell death. Receptor of advanced glycation end-products (RAGE), is a major receptor that facilitates the toxicity of Aβ [75], and Aβ1-42 mediates abnormal autophagy through the RAGE-associated pathway [75]. Treatment with Aβ peptide induces dysfunctional autophagy in astrocytes where p62 is aggregated and LC3-I/LC3-II transformation is decreased [76]. Mitochondrial abnormalities are caused by Aβ-mediated dysfunction of the voltage-dependent anion channel 1 protein (VDAC1) as well as dynamin-related protein 1 (Drp1) [77]. Mitophagy promotes the removal of injured mitochondria, where PTEN-induced putative kinase 1 (PINK1) plays a vital role in regulating mitochondrial function [78]. Lower levels of PINK1 have been linked to AD pathology. Aβ pathology, which promotes cognitive and synaptic dysfunction, was alleviated by PINK1-mediated mitophagy [79]. Interestingly, overexpression of PINK1 enhanced the removal of injured mitochondria through upholding mitophagy in AD. In a mouse model of AD, hippocampal Aβ decreases PINK1 expression, which reduces mitophagy and causes cognitive decay [9].

## 6. Therapeutic Action of APP Triggered by Autophagy

The incidence of AD has posed a global health burden on elderly people and is predicted to increase significantly worldwide. Considerable effort has been devoted to developing drugs for the treatment of AD, focusing on drug structures and potential molecular mechanisms of AD. An example of a generally accepted hypothesis is that reduced levels of acetylcholine causes AD in neurons. However, drugs targeting acetylcholine, the so called “cholinesterase inhibitors” could poorly improve AD [80,81]. Therefore, examining other drugs, including potential autophagy regulators might have a greater potential for the treatment of AD.

### 6.1. Use of Small Molecules to Modulate Autophagy in AD

The hallmark of AD pathogenesis is the accumulation of amyloid beta proteins and hyperphosphorylated tau, which are considered toxic to neurons [82]. Dysregulated or insufficient autophagy might be causative factors behind the development and progression of AD [21], and therefore, the discovery of drugs targeting autophagy-related signaling pathways might be an attractive approach for the treatment and management of AD [17]. However, the Autophagy Small Molecule Database (AutophagySMDB) comprising small molecules is growing gradually. Further, an extensive database containing numerous target proteins of small molecule modulators that have curated indirect or direct evidence will be completed in the near future [83]. Moreover, the significance of autophagy in numerous disease states, in addition to requiring deeper examination of its molecular mechanisms. Kuang et al. has been revealed that several small molecule autophagy modulators, which may serve as prospective tools for therapeutics of AD [84] (Figure 6). Sirtuin1 (SIRT1), a positive regulator of autophagy, increases the expression levels of Atg5, Beclin-1, and LC3-II and accelerates the clearance of Aβ [85]. In addition, PPARα-mediated activation of autophagy facilitated the clearance of APP and decreased Aβ pathology in APP/PS1 mice [86]. Treatment with PPARα agonists decreased Aβ levels in the hippocampus and cortex, and improved autophagosome biogenesis. Together, these observations suggest that PPARα is a critical player involved in autophagy during Aβ processing [87]. Recently, it has been found that LC3-associated endocytosis assists Aβ clearance in addition to alleviates murine AD [88]. Spleen tyrosine kinase (SYK) inhibits mTOR signaling with attenuation of tau accumulation and reduces loss of synaptic function in vitro and in vivo [89]. Small molecule estrogen receptor β (ER) has been found to promote neuroprotective and tau degradative activity via LC3-II and Atg7 enhancement of extracellular Aβ1-42 degradation via the autophagy-lysosome system in AD [90]. Orientin increases learning and memory function in addition to increase clearance of Aβ via the autophagic pathway of LC3-II upregulation as well as p62 and cathepsin D degradation in an AD mouse model [91]. Additionally, transmembrane p24 trafficking protein 10 (TMED10) has been described to activate autophagy via ATG4B activation through decreaseing Aβ production in AD patients [92].

### 6.2. Use of Natural Compounds to Modulate Autophagy in AD

Natural products have been used to treat several neurodegenerative disorders and cancer, and have been targeted as autophagy inducers [93]. Recent studies have revealed that active compounds in natural products exhibit curative effects against AD via various mechanisms, including anti-cholinesterase activity, anti-apoptosis, and neuroprotective effects via anti-oxidation through targeting autophagy [94]. Emerging evidence suggests that natural compounds are attractive sources of autophagy regulators [95]. Several reports demonstrate that the active compounds regulating autophagy pave a new therapeutic approach for neurodegenerative diseases [17]. Examples of plant-derived active components that ameliorate the symptoms of AD by targeting autophagy are summarized in Table 1.

Alkaloids are important examples of active compounds isolated from plants and show anti-cholinesterase and modulatory effects on autophagy and are implicated in the treatment of neurodegenerative diseases. For instance, some alkaloids can modulate autophagy in AD [96], such as *Dendrobium nobile* Lindl (DNLA) extracts possess alkaloid components that are capable of hindering axonal degeneration [97]. Extra-Virgin Olive Oil (EVOO) reduces neuroinflammation through activation of autophagy by AMPK-ULK1 pathway in TgSwDI mice model [98]. Plant alkaloid berberine ameliorated learning and memory functions and accelerated Aβ clearance in a mouse model of AD [99]. Berberine can also promote autophagy in the brain [100] and has neuroprotective activity [101]. An oxindole alkaloid, corynoxine, obtained from *Uncaria rhynchophylla* (Miq.) is another example of an autophagy enhancer [102]. Moreover, an isomer of corynoxine, corynoxine B, was shown to promote autophagy and reduce the accumulation of Aβ by facilitating the degradation of APP [103].

Many studies have demonstrated that active components of flavonoids can affect autophagy in various diseases. The plant flavonoid silibinin, extracted from *Silybum marianum* ameliorated Aβ1-42-induced depression in rats, and alleviated neuronal damage in the hippocampus by inhibiting autophagy [107]. Another flavonoid component, wogonin, isolated from *Scutellaria baicalensis*, enhances Aβ clearance in cortical astrocytes and reduces Aβ deposition by modulating autophagy [114]. Moreover, hesperetin and its glycoside hesperidin are implicated in the protection of neurons by decreasing Aβ-mediated autophagy [112].

A variety of products isolated from *Panax* ginseng have also been shown to provide neuroprotection and ameliorate memory function in dementia [124]. For example, Rg2 ginseng triggers autophagy [104], accelerates the clearance of aggregated proteins, decreases the accumulation of cerebral Aβ, and ameliorates cognitive functions via autophagy in a mouse model of AD [103]. Protopanaxadiol is associated with axonal outgrowth in neuronal degeneration and can ameliorate memory disorders in AD mice [105]. DDPU has been implicated in the development of AD behavior associated with autophagy [105]. Another alkaloid gypenoside XVII found in ginseng *Panax notoginseng* enhanced the removal of amyloid deposition in the hippocampus and cortex of mice, and it also exerts a neuroprotective effect in AD by triggering autophagy [108]. Madecassoside, a triterpenoid saponin compound, inhibits autophagy by increasing the levels of Bcl-2 and decreasing Beclin-1 in neuronal cells [111]. It has also been observed that madecassoside improves cognitive function and synaptic plasticity in a mouse model of AD [125]. However, modulation of O-GlcNAcylation regulates autophagy in astrocytes and controls the antidepressant-like phenotype in neurons [126,127]. Gintonin has been used for the treatment or prevention of AD through elevation of hippocampal neurogenesis in APPswe/PSEN-1 double Tg mouse model of AD [122]. It has been demonstrated that gintonin-mediated treatment is nontoxic and possibly beneficial in cognitively impaired elderly patients with AD [123].

Curcumin relieves cognitive impairment and inhibits Aβ formation by blocking autophagy through the PI3K/Akt/mTOR pathway [115]. The polyphenol resveratrol is widely considered as a therapeutic because of its beneficial effects observed in AD models [128]. Recently, it has been found that decreasing the production of Aβ hinders the development of AD. The synthesis of Aβ was reduced in cells by modulating signaling pathways such as triggering the AMPK pathway and preventing mTOR to activate autophagy. It was also reported that oral administration of resveratrol suppressed Aβ accumulation in the cortex [129]. One of the primary bioactive components, tetrahydroxystilbene-2-O-glycoside from Radix *Polygoni multiflori*, decreases APP expression and enhances cognitive activity in transgenic mice with AD by reducing Beclin-1 and LC3-II expression through autophagy [103]. Emodin, obtained from *Rheum palmatum* L., reduces LC3-II expression and increases Bcl-2 expression, and has been used to treat AD [103]. The polyphenolic compound, carnosic acid from *Rosemarinus officinalis* is involved in the reduction of neurotoxicity induced by amyloid deposition, and decreases the accumulation of amyloid aggregates and hyperphosphorylation of tau [119]. Consumption of Ginkgo biloba extract enhanced the cognitive and synaptic function in a mouse model of AD by partially activating autophagy [109]. In addition, some compounds with special structural features, such as arctigenin, tripchlorolide, and β-asarone were shown to improve memory by modulating autophagy-related signaling pathways. Significant levels of arctigenin were observed in the brain, indicating that it might cross the blood-brain barrier [130]. Therefore, natural products could be considered as therapeutic targets for the management of AD.

### 6.3. Use of FDA-Approved Drugs to Modulate Autophagy in AD

FDA-preapproved as well as approved drugs have the main advantage of being characterized fully for their pharmacological activity. These kinds of drugs have been shown to be safe in reducing toxicity, possibility of oral administration, ability to cross the BBB, and half-life are contained within pharmacologically standard. Clomipramine, an FDA-approved drug used for the treatment of psychiatric disorders, was shown to block the fusion of autophagosomes with the lysosomes, thus interfering with autophagic flux [131]. Another FDA-approved drug, a benzoporphyrin derivative, known as verteporfin, is likely to inhibit the formation of autophagosomes in the presence of chloroquine (CQ), a lysosomal inhibitor [132]. APP 5′ UTR-directed drugs decreased APP levels in SH-SY5Y neuroblastoma cells [133]. As shown in Figure 7, lower levels of Aβ peptides were achieved by using FDA-preapproved drugs that lowered intracellular APP holoprotein levels in SH-SY5Y cells have been demonstrated by Morse et al. [134]. The pharmacological action of DMP, DFO, paroxetine, phenserine, and tetrathiolmobdylate in decreasing the levels of APP and Aβ peptide is shown in Figure 7. Azithromycin dramatically changed the processing of APP [135]. Thus, it has been reported that a subsection of drugs that have been selected to stimulate APP 5′ UTR-mediated translation is, in addition, coactivators of the non-amyloidogenic pathway of APP processing. Therefore, it is urgently required to test an increased number as well as more sophisticated FDA-approved drugs with relative effectiveness in larger group of animals to modulate autophagy in AD pathogenesis.

## 7. Conclusions

From the above discussion, it has been suggested that autophagy is an important cellular process in AD pathogenesis, which regulates the production and degradation of primary pathological proteins, such as Aβ and p-tau. While autophagy plays a dual role in AD, systematic research with new evidence will provide information on the modulation of autophagy in AD treatment. Taken together, recent studies have demonstrated that a variety of bioactive components from medicinal herbs, small molecules, and FDA-approved drugs are likely to improve AD via modulation of autophagy.

## Figures and Tables

**Figure 1 biomedicines-09-00005-f001:**
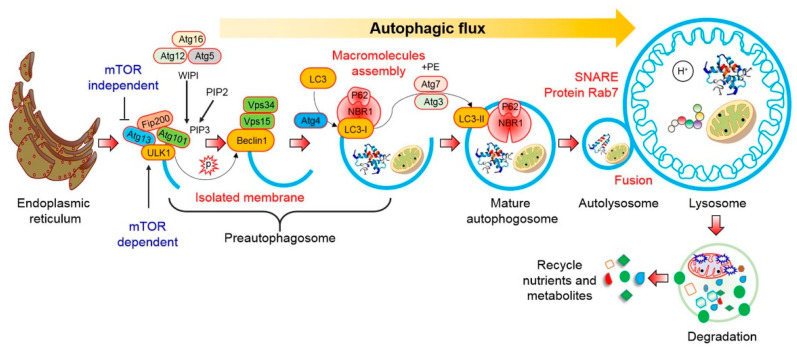
Schematic diagram of different steps of autophagy. Autophagy is initiated by inactivation of the mTOR pathway and induction of numerous autophagy-related proteins. Pre-autophagosome synthesis involves the coupling of LC3 with ULK1 and Vps34 proteins. Importantly, AMPK-mTOR signaling are involved to initiate pre-autophagosome or phagosphore formation. Sometimes, mTOR independent pathways are associated to initiate phagosphore formation. Subsequently, the pre-autophagosome forms the double membrane vesicle that sequesters cytoplasmic contents and forms the mature autophagosome. Finally, the mature autophagosomes fuse with the lysosomes wherein the contents are degraded and recycled back into the cytoplasm which may further be used as nutrients.

**Figure 2 biomedicines-09-00005-f002:**
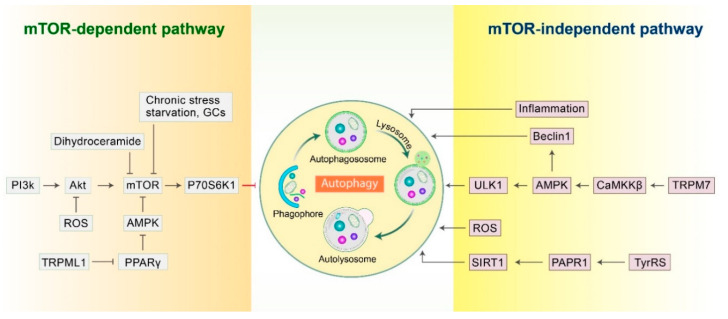
Autophagy regulation by the mTOR signaling pathway. Autophagy can be regulated by both mTOR-dependent as well as mTOR-independent pathways. mTOR phosphorylation can lead to ribosomal P70S6K1 phosphorylation, which is an mTOR substrate protein, thereby preventing autophagy initiation. Autophagy can additionally be stimulated by certain other elements for example starvation, chronic stress, as well as GCs through mTOR inhibition. Furthermore, TRPML1/PPARγ/AMPK/mTOR and PI3K/Akt/mTOR are positive and negative autophagy regulators, respectively, and stimulation as well as prevention of these pathways may trigger autophagy. Also, dihydroceramide is a regulator of mTOR-mediated autophagy induction. However, ROS stimulates autophagy via mTOR-dependent (inhibit Akt) as well as mTOR-independent pathways directly activate autophagy. Besides, inflammatory stimulation of microglia plays a role in autophagy initiation. TRPM7, CaMKKβ, AMPK as well as TyrRS, PARP1, and SIRT1 are the most important positive modulators in mTOR-independent autophagy regulation.

**Figure 3 biomedicines-09-00005-f003:**
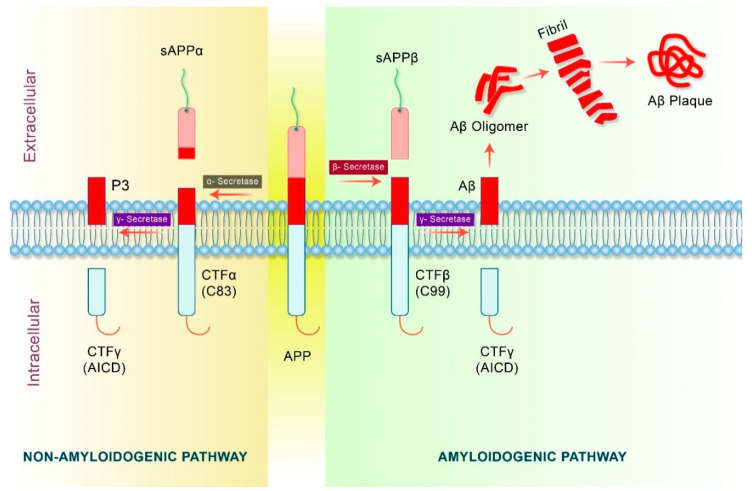
APP processing pathways in AD. APP are generally processing in two different pathway, amyloidogenic and non-amyloidogenic. In the non-amyloidogenic pathway, APP is cleaved by α-secretase to form two fragments, an intracellular C-terminal fragment α (CTFα), C83, and an extracellular fragment, soluble amyloid precursor protein α (sAPPα). Cleavage of C83 fragment via γ-secretase yields a short peptide fragment, P3, and an APP intracellular domain (AICD). In the amyloidogenic pathway, APP is cleaved by β-secretase which produces a soluble amyloid precursor protein β (sAPP β) and a C-terminal fragment β (CTFβ) or C99 fragment. C99 fragment is cleaved by γ-secretase to produce Aβ and C-terminal fragment γ (CTFγ) or AICD. Aβ further forms Aβ oligomers which subsequently results in the formation of fibrils and neurotoxic Aβ plaques extracellularly.

**Figure 4 biomedicines-09-00005-f004:**
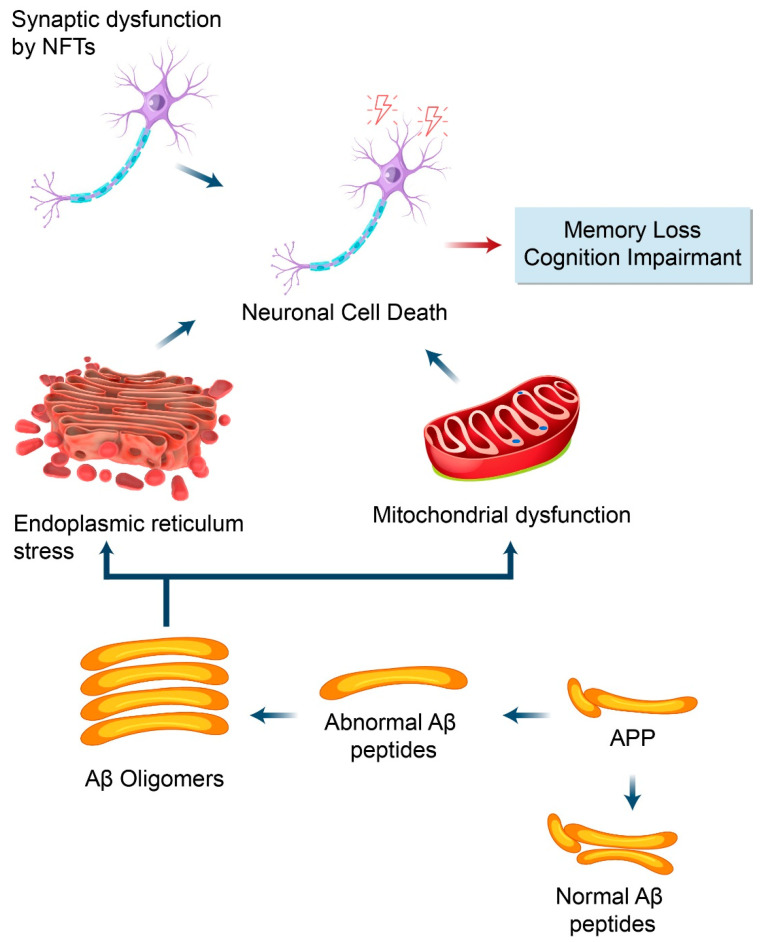
Schematic representing modulation of AD pathogenesis by APP. AD pathogenesis is initiated upon the accumulation of abnormal Aβ peptides ((1-42 Aβ peptide of 36–43 amino acids) derived from APP proteolytic cleavage through γ-secretase and β-secretase. Aβ oligomers stimulate ER stress and trigger mitochondrial dysfunction (generally produce ROS) which lead to cause neuronal death and impairment cognition function during AD progression. In addition, synaptic dysfunctions are also initiated due to hyperphosphorylation of tau which leads to the formation of neurofibrillary tangles (NFTs) which causes synaptic dysfunction and neuronal loss in AD pathogenesis.

**Figure 5 biomedicines-09-00005-f005:**
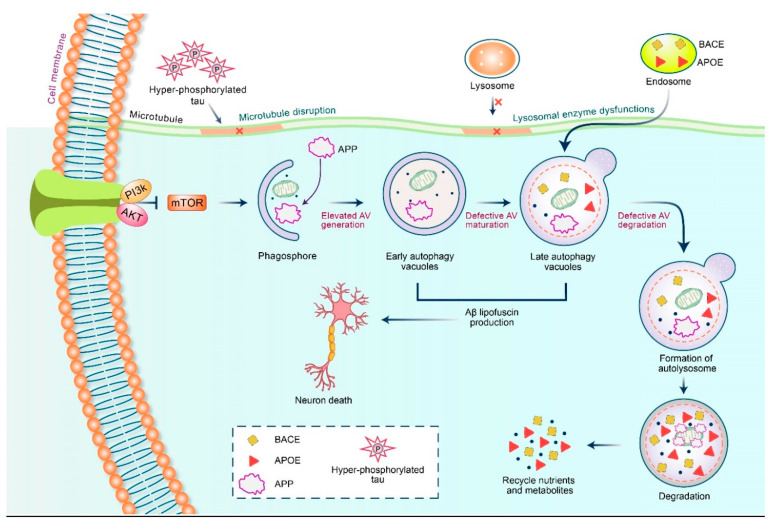
APP, BACE, and ApoE are AD-related molecules and eliminated by autophagy. In the initial stage of AD, autophagic vacuoles (AVs) are formed due to the stress induced by APP mutants which ultimately damage the mitochondria. During the late stage of AD, maturation as well as degradation of autophagosomes are blocked by microtubule disruption which causes hyperphosphorylated tau accumulation. Tau hyperphosphorylation might inhibit microtubule assembly in addition to disrupt the preassembled microtubules. Eventually, dysfunction of lysosomal enzymes interferes with autophagosome-lysosome fusion in AD. Taken together, these defects of autophagy contribute to accumulation of AVs along with other AD-related molecules which increased intracellular Aβ deposition as well as lipofuscin thereby causing neuronal cell degeneration and death.

**Figure 6 biomedicines-09-00005-f006:**
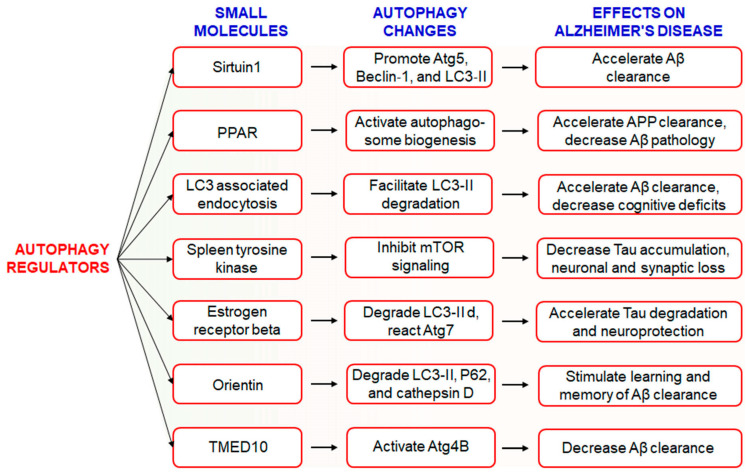
Implication of several small molecules that modulate autophagy and their mode of action in AD treatment. It has been mentioned that a single drug may target multiple factors that might be involved in diverse pathological conditions thereby augmenting treatment efficacy. Additionally, it has been limited negative features of a conventional single-target drug and combination drugs as well. All the small molecules are used to treat AD via modulation several autophagic signaling.

**Figure 7 biomedicines-09-00005-f007:**
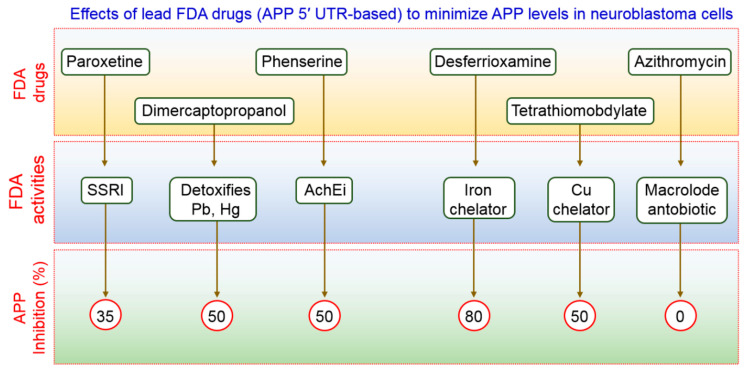
Effects on FDA-approved drugs used for the treatment of AD-associated disorders. The therapeutic action of these FDA-approved compounds has been demonstrated as proof of concept in vivo for selectively reducing APP expression in AD. All the drugs are shown to reduce APP inhibition.

**Table 1 biomedicines-09-00005-t001:** Modulation of autophagy by natural products and their therapeutic implication in Alzheimer’s disease.

Natural Products	AD Model	Activities/Effects	Molecular Mechanism	References
Dendrobium nobile Lindl alkaloid, DNLA	Hippocampus neurons of Aβ25-35	Protective effects of axonal degeneration	Autophagic flux enhancement	[97]
Extra-Virgin Olive Oil (EVOO)	TgSwDI mice	Neuroinflammation reduction	AMPK-ULK1 pathway induction	[98]
Ginsenoside Rg2	5×FAD transgenic mice	Removal of Aβ aggregation	AMPK/ULK1-mediated autophagy induction	[104]
Protopanaxadiol derivative DDPU	APP/PS1 mice model	Stimulates the clearance of Aβ	Inhibition of PI3K/mTOR-mediated autophagy induction	[105]
Berberine	3×Tg-AD mice	Promotes the clearance of Aβ	Activates Bcl2/Beclin1-mediated autophagy induction	[101,106]
Flavonoids Silibinin	Aβ1-42-induced rat model	Attenuates neuronal damage	Inhibits autophagy	[107]
Corynoxine B	Tg2567 mice, N2a-SwedAPP cell model	Augments APP and Aβ degradation	Pathway that induces autophagy is unknown	[103]
Gypenoside XVII	APP/PS1 transgenic mice	Prevents Aβ accumulation	Promotes TFEB to induce autophagy	[108]
*Ginkgo biloba* extract	TgCRND8 mice	Improves cognitive function	Induces autophagy	[109]
*Radix polygalae* extract	Cell model of CHO-APP/BACE1	Decreases Aβ1-40 levels	Activates AMPK/mTOR and promotes autophagy	[110]
Madecassoside	D-galactose-induced mouse model	Autophagy inhibition	Increases Bcl-2 and decreases Beclin-1	[111]
Hesperetin	N2a cell model	Increases Aβdamage	Autophagy inhibition	[112]
*Morus alba* extract	SH-SY5Y cells	Autophagy induction	mTOR-dependent autophagy pathway	[94,113]
Wogonin	SH-SY5Y-APP primary cortical astrocytes	Enhances Aβ removal	Activates ULK1/mTOR and induces autophagy	[114]
Curcumin	APP/PS1 transgenic mice	Prevents Aβ deposition	Inhibits PI3K/mTOR and induces autophagy	[115]
Resveratrol	N2a-APP cells, HEK293-APP cells	Decreases Aβ production and aggregation	Induces autophagy by activating AMPK/mTOR signaling	[116,117]
Sulforaphane	AD model	Nrf2 signaling	Induces autophagy	[118]
Carnosic acid	Aβ25-35-induced SHSY5Y cells	Inhibition of Aβ1-42 aggregation	Activates AMPK/mTOR and induces autophagy	[119]
Tripchlorolide	5×FAD transgenicmice	Reduces cerebral Aβ deposits	Activates PI3K/mTOR pathway	[120]
β-asarone	APP/PS1 transgenic mice	Decreases Aβ level	Activates PI3K/mTOR and inhibits autophagy	[121]
Oxyresveratrol	SH-SY5Y cell model	Stimulates autophagy	Atg5/7, Beclin-1, and LC-3 induction	[31]
18α-Glycyrrhetinic acid	SH-SY5Y cell model	Induction of autophagy flux	mTOR-dependent autophagy induction	[30]
Gintonin	Mouse cortical Astrocytes, APPswe/PSEN-1	Autophagic flux induction, cognition improvements	Beclin-1, Atg5/7, LAMP-1 induction, elevation of hippocampal neurogenesis	[29,122,123]
Emodin	APP/PS1 mice	Autophagy inhibition	Activates Bcl-2/Beclin-1/PIK3C3 pathway	[103]

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
