# Peer review of "Modulatory Effects of Autophagy on APP Processing as a Potential Treatment Target for Alzheimer’s Disease"

_biomedicines, 2020, doi:10.3390/biomedicines9010005_

Round 1

Reviewer 1 Report

The authors developed an interesting review on the role of autophagy on APP processing. The subject is a hot topic and deserves attention. References are up to date, figures are useful to better understand the cited mechanisms and pathways, and table 1 adds to the value of the review.

I only have some remarks concerning the form and a few advices.

line 27: a pathology is either chronic or is not. "Most chronic" does not make sense.

30: 70% is a pretty precise value. Can the authors add other references validating this number?(possibly in open access).

55: leads

59: hyperphosphorilated tau is not an AD form, it is a hallmark.

63: the subject is "neurotoxic effects", so ...are hallmarks

75: aggregates

116: comprised of

171: results

230: the majority of APP

Author Response

The authors developed an interesting review on the role of autophagy on APP processing. The subject is a hot topic and deserves attention. References are up to date, figures are useful to better understand the cited mechanisms and pathways, and table 1 adds to the value of the review.

I only have some remarks concerning the form and a few advices.

First of all, we would like to express our sincere gratitude for the time and effort the reviewer had put into reviewing our manuscript.

line 27: a pathology is either chronic or is not. "Most chronic" does not make sense.

>>Response: We checked and modified the sentence (page 3, line 57).

30: 70% is a pretty precise value. Can the authors add other references validating this number?(possibly in open access).

>>Response: We added more references (page 3, line 62).

55: leads

>>Response: We corrected accordingly (page 4, line 83).

59: hyperphosphorilated tau is not an AD form, it is a hallmark.

>>Response: We corrected the sentence (page 4, line 87-88).

63: the subject is "neurotoxic effects", so ...are hallmarks

>>Response: We corrected the sentence according to reviewer suggestion (page 4, line 92).

75: aggregates

>>Response: We corrected the word (page 5, line 115).

116: comprised of

>>Response: We corrected the word accordingly (page 6, line 159).

171: results

>>Response: We corrected the word (page 9, line 215).

230: the majority of APP

>>Response: This information has been deleted suggesting by reviewer 3.

Reviewer 2 Report

The article titled "Promising modulatory effects of autophagy on APP processing as a potential treatment for Alzheimer's disease" is a well written comprehensive review in which authors first discussed both the APP Protein processing in Alzheimer's disease and autophagy pathways separately, and then elaborated on the interaction of both these pathways. Small molecules that regulate autophagy and thereby the pathogenesis of Alzheimer's disease are also discussed. The Manuscript is well organized.

The only concern I have is language. The abstract is poorly written and didn't highlight the main features of the review.

Also, please avoid the usage of lengthy sentences and improve the languages in the abstract and also where ever possible.

Minor suggestions:

  • line 17, a bidirectional link "between" instead "to"
  • line19, dynamic modification? or modulation?
  • Figure legends need to be explained more clearly

Author Response

The article titled "Promising modulatory effects of autophagy on APP processing as a potential treatment for Alzheimer's disease" is a well written comprehensive review in which authors first discussed both the APP Protein processing in Alzheimer's disease and autophagy pathways separately, and then elaborated on the interaction of both these pathways. Small molecules that regulate autophagy and thereby the pathogenesis of Alzheimer's disease are also discussed. The Manuscript is well organized.

First of all, we would like to express our sincere gratitude for the time and effort the reviewer had put into reviewing our manuscript.  

The only concern I have is language. The abstract is poorly written and didn't highlight the main features of the review.

>>Response: We modified the abstract with highlighting the main features of the review (page 2, line 33-49)

Also, please avoid the usage of lengthy sentences and improve the languages in the abstract and also where ever possible.

>>Response: We checked and improved the languages quality with avoiding usage of lengthy sentences.

Minor suggestions:

line 17, a bidirectional link "between" instead "to"

>>Response: We corrected it accordingly (page 2, line 46).

line19, dynamic modification? or modulation?

 >>Response: we used modulation according to the reviewer suggestion (page 2, line 48).

Figure legends need to be explained more clearly

>>Response: We checked and added more information in each figure legends.

Reviewer 3 Report

This manuscript reviews mechanisms responsible for the role of autophagy in APP expression and processing with a focus on drug treatments. This is an important topic. The presentation raised several questions and comments that must be addressed in a revision.

Major concerns:

1) In general in this review article, the references have to be sourced from the original paper, not a summary of review articles, for examples: reference #31 on page 6, line 140; reference #33 on page 6, line 141. Original ideas of papers should be cited. Of course as a service to your readers you can cite the review as "an good overview of recent discussions/research can be found in....".

2) It appears that the self-citation in this review article is a bit excessive. About 25 of total 133 references (≈20%) are from their own work or that of colleagues. The authors should consider balanced citations and cite similar works of others groups.

3) You have not said much about cellular localization of APP processing in the introduction (page2). For example, Aβ peptides are generated within intracellular organelles such as the ER, but also extracellularly. AβPP is folded and modified in the ER and transported through the Golgi complex to the plasma membrane. It was proposed that Aβ oligomers accumulate in the ER lumen as a result of deficits in axonal transport (Umeda et al. J Neurosci Res. 2011; 89:1031-42). The authors need to expand this section to include the important physiological function.

4) There are several repetitive sentences with same information throughout of the manuscript and should be removed, for example:

a) Page 2, lines 42-44 (APP is taken up…). This is already mentioned on page 3.

b) Page 2, lines 53-54 (due to mutations in genes encoding APP, PSES1, and PSEN2, and). This was already explained in line 51.

c) Section 5 on page 11: The information presented in this section is already mentioned elsewhere in other sections and hence section 5 should be deleted. Other option would be to combine sections 4 and 5 and avoid repetitive information.

d) Page 13, lines 262-263: Remove this sentence as it is already explained on page 4.

5) Page 6, lines 131-133: In my understanding is that ceramide (generated from dihydroceramide) activates mTOR1 and thereby inhibits autophagy. Is this so, than an arrow should be shown in in signaling pathway presented in Figure 2?

6) The section 7.1 (page 16) is too short and doesn’t describe in detail the figure 6. The authors need to expand this section and include original references. In addition, dendrobium nobile lindl alkaloid and extra virgin olive oil are considered as natural products and should be moved to section 7.2. Moreover, why are other small molecules described elsewhere in the manuscript such as Sirtuin1 or PPAR not included in this schematic diagram?

Author Response

This manuscript reviews mechanisms responsible for the role of autophagy in APP expression and processing with a focus on drug treatments. This is an important topic. The presentation raised several questions and comments that must be addressed in a revision.

First of all, we would like to express our sincere gratitude for the time and effort the reviewer had put into reviewing our manuscript.  

Major concerns:

1) In general in this review article, the references have to be sourced from the original paper, not a summary of review articles, for examples: reference #31 on page 6, line 140; reference #33 on page 6, line 141. Original ideas of papers should be cited. Of course as a service to your readers you can cite the review as "an good overview of recent discussions/research can be found in....".

 >>Response: We replaced the reference by original paper (page 7, line 184, reference#36), (page 7, line 185, reference#38).

2) It appears that the self-citation in this review article is a bit excessive. About 25 of total 133 references (≈20%) are from their own work or that of colleagues. The authors should consider balanced citations and cite similar works of others groups.

 >>Response: We are grateful the reviewer for this important points. We reduced our own references and used as much as related to this paper (total#15).

3) You have not said much about cellular localization of APP processing in the introduction (page2). For example, Aβ peptides are generated within intracellular organelles such as the ER, but also extracellularly. AβPP is folded and modified in the ER and transported through the Golgi complex to the plasma membrane. It was proposed that Aβ oligomers accumulate in the ER lumen as a result of deficits in axonal transport (Umeda et al. J Neurosci Res. 2011; 89:1031-42). The authors need to expand this section to include the important physiological function.

 >>Response: We added one more paragraph in introduction about cellular localization and  important physiological function of APP (page 4, line 94-105).

4) There are several repetitive sentences with same information throughout of the manuscript and should be removed, for example:

  1. a) Page 2, lines 42-44 (APP is taken up…). This is already mentioned on page 3.

>>Response: we deleted the this repetition (page 4, line 111-112).

  1. b) Page 2, lines 53-54 (due to mutations in genes encoding APP, PSES1, and PSEN2, and). This was already explained in line 51.

>>Response: We reorganized these two sentences and make one sentence (page 3, line 82-84).

  1. c) Section 5 on page 11: The information presented in this section is already mentioned elsewhere in other sections and hence section 5 should be deleted. Other option would be to combine sections 4 and 5 and avoid repetitive information.

>>Response: According to the reviewer suggestion we deleted some part of section 5 and combined with section avoiding repetitive information (page 10, line 273-284).

  1. d) Page 13, lines 262-263: Remove this sentence as it is already explained on page 4.

 >>Response: We deleted the sentence (page 13, line 296).

5) Page 6, lines 131-133: In my understanding is that ceramide (generated from dihydroceramide) activates mTOR1 and thereby inhibits autophagy. Is this so, than an arrow should be shown in in signaling pathway presented in Figure 2?

 >>Response: Previously, it has been found that dihydroceramide desaturases activates mTOR and inhibit autophagy. Later, it is found that dihydroceramide decreases mTOR and promote autophagy. Therefore, we modified the paragraph according to figure 2 and did not shown arrow in figure 2. Additionally, we replaced figure 2 by a new figure. (page 7, line 173-178, figure 2).

6) The section 7.1 (page 16) is too short and doesn’t describe in detail the figure 6. The authors need to expand this section and include original references. In addition, dendrobium nobile lindl alkaloid and extra virgin olive oil are considered as natural products and should be moved to section 7.2. Moreover, why are other small molecules described elsewhere in the manuscript such as Sirtuin1 or PPAR not included in this schematic diagram?

>>Response: We described all the contents with original references (page 16, line 382-397, figure 6).

We remove dendrobium nobile lindl alkaloid and extra virgin olive oil from figure 6 and move to text in section 6.2 and table 1 (Page 18, line 317-421, table 1).

Moreover, small molecules Sirtuin1 and PPAR are included in schematic diagram in figure 6 and contents are described in section 6.1 (Page 16, line 332-388, figure 6).

Round 2

Reviewer 3 Report

The authors have addressed the reviewer's concerns and the manuscript is improved. Here is one minor suggestion:

I find that the introduction is a bit too long. Therefore I am suggesting to move the second paragraph on page 4 to page 9 as this content fits better in section 3 "Neuronal roles of APP".

Author Response

The authors have addressed the reviewer's concerns and the manuscript is improved. Here is one minor suggestion:

First of all, we would like to express our sincere gratitude for the time and effort the reviewer had put into reviewing our manuscript.

I find that the introduction is a bit too long. Therefore I am suggesting to move the second paragraph on page 4 to page 9 as this content fits better in section 3 "Neuronal roles of APP".

>>Response

As the reviewer suggestion, we move second paragraph from introduction section to section 3 "Neuronal roles of APP" (page 9, line 214-225).

This manuscript is a resubmission of an earlier submission. The following is a list of the peer review reports and author responses from that submission.

Round 1

Reviewer 1 Report

In the manuscript authors made a collection of several molecules that have been described to modify autophagy and have an effect on neurodegeneration caused by amyloid accumulation in Alzheimer’s disease models, both in vitro and in vivo

Unfortunately, there are some errors in basic concepts that must been deeply analyzed. Here I summarize some of them.

The introduction should begin with some information about Alzheimer’s disease that is absent at all. Authors should also mention that the most of the cases are not related to any mutation, are sporadic

I suppose authors are talking about macroautophagy ??

Line 69 what are irregular peptides?

Lines 74-75 NFTs are formed by protein tau in a hyper-phosphorylated form.

Line 88 what are pathological amyloids? Maybe a reference is missing near here

Lines 143-145. Reference 25 described an effect of Dihydroceramide Desaturase 1 Inhibitors on autophagy by p62, S6 and LC3 levels, as well as autophagy flux, but no analyze of mTOR was performed

Lines180-182 “…amyloid formation is not an intended physiological function…”  Reference is missing and the meaning is a mistake or not clear

Lines 204-223 The explanation of APP cleavage is so confusing. Authors mixed amyloidogenic and non-amyloidogenic pathways in the text. I suggest to rebuild this section in two separated parts for a better understanding

Line 213 would be alpha-secretase instead of gamma-secretase?

Figure 3. Beta-secretase cleavage APP in the transmembrane domain. Amyloid peptide image is wrong.  Authors can check reference 50 to remodel the figure.

Line 232 What is ‘regular condition’?

Line 233 peptides are non-amyloidogenic?

Line 236 The options would be alpha or beta-secretase for non-amyloidogenic or amyloidogenic pathway. Gamma-secreatase is present in both of them.

Line 239 amyloid peptide is accumulated extracellularly (not intracellularly) as plaques (not plagues)

Line 241 there are several studies that study the relationship between the accumulation of amyloid peptide and the increase in phosphorylated state of tau protein

Figure 4. What are abnormal Abeta peptides?

Figure 5 legend. Authors mention related molecule ApoE that doesn’t appear anywhere. What is the stress of mutant APP? The following text with the reference maybe out of the legend. The legend should explain what we can see in the figure are this explanation is nowhere

Line 299-300. The sentence is not correct (50)

Line 331 treiggring

Figure 6 There is no mention in the text about this figure and the references are missing

Lines 351-352 confusing

Table 1 and text: Authors mixed products that inhibit or activate authophagy, however all of them reduce Abeta. There is no discussion about this point or any other. This is just a compilation

Reviewer 2 Report

This paper could be of interest to readers. However, its organizaction, frequent repetitions, poor English all disqualify this submission. I think to many authors cause the paper to be chaotic, of uneven quality and uncomprehensible. This paper has to be rewritten and can be than resubmitted. It needs native English speaker to correct the language.